# Presence of Infection by *Mycobacterium avium* subsp. *paratuberculosis* in the Blood of Patients with Crohn’s Disease and Control Subjects Shown by Multiple Laboratory Culture and Antibody Methods

**DOI:** 10.3390/microorganisms8122054

**Published:** 2020-12-21

**Authors:** J. Todd Kuenstner, Raghava Potula, Tim J. Bull, Irene R. Grant, Antonio Foddai, Saleh A. Naser, Horacio Bach, Peilin Zhang, Daohai Yu, Xiaoning Lu, Ira Shafran

**Affiliations:** 1Pathology and Laboratory Medicine, Lewis Katz School of Medicine, Philadelphia, PA 19140, USA; toddkuenstner@gmail.com; 2Center for Substance Abuse, Lewis Katz School of Medicine, Philadelphia, PA 19140, USA; 3Institute for Infection & Immunity, St. George’s, University of London, London SW17 0RE, UK; tbull@sgul.ac.uk; 4Institute for Global Food Security, School of Biological Sciences, Queen’s University, Belfast BT9 5DL, UK; i.grant@qub.ac.uk (I.R.G.); antonio.foddai@abdn.ac.uk (A.F.); 5Burnett School of Biomedical Sciences, College of Medicine, University of Central Florida, Orlando, FL 32816, USA; Saleh.Naser@ucf.edu; 6Division of Infectious Diseases, Faculty of Medicine, The University of British Columbia, Vancouver, BC V6H 3Z6, Canada; horacio.bach@gmail.com; 7PZM Diagnostics, Charleston, WV 25301, USA; pzmllc@gmail.com; 8Department of Clinical Sciences, Lewis Katz School of Medicine, Philadelphia, PA 19140, USA; dyu@temple.edu (D.Y.); xiaoning.lu@temple.edu (X.L.); 9Shafran Gastroenterology Center, Winter Park, FL 32789, USA; shafranira@gmail.com

**Keywords:** MAP, Crohn’s Disease, bacteremia, IBD, autoimmune disease

## Abstract

*Mycobacterium avium* subspecies *paratuberculosis* (MAP) has long been suspected to be involved in the etiology of Crohn’s disease (CD). An obligate intracellular pathogen, MAP persists and influences host macrophages. The primary goals of this study were to test new rapid culture methods for MAP in human subjects and to assess the degree of viable culturable MAP bacteremia in CD patients compared to controls. A secondary goal was to compare the efficacy of three culture methods plus a phage assay and four antibody assays performed in separate laboratories, to detect MAP from the parallel samples. Culture and serological MAP testing was performed blind on whole blood samples obtained from 201 subjects including 61 CD patients (two of the patients with CD had concurrent ulcerative colitis (UC)) and 140 non-CD controls (14 patients in this group had UC only). Viable MAP bacteremia was detected in a significant number of study subjects across all groups. This included Pozzato culture (124/201 or 62% of all subjects, 35/61 or 57% of CD patients), Phage assay (113/201 or 56% of all subjects, 28/61 or 46% of CD patients), TiKa culture (64/201 or 32% of all subjects, 22/61 or 36% of CD patients) and MGIT culture (36/201 or 18% of all subjects, 15/61 or 25% of CD patients). A link between MAP detection and CD was observed with MGIT culture and one of the antibody methods (Hsp65) confirming previous studies. Other detection methods showed no association between any of the groups tested. Nine subjects with a positive Phage assay (4/9) or MAP culture (5/9) were again positive with the Phage assay one year later. This study highlights viable MAP bacteremia is widespread in the study population including CD patients, those with other autoimmune conditions and asymptomatic healthy subjects.

## 1. Introduction

*Mycobacterium avium* subspecies *paratuberculosis* (MAP) is accepted as the cause of Johne’s Disease (JD) [1], a chronic diarrheal wasting disease of cattle and a wasting disease in sheep and goats [2] and has long been suspected to be involved in the etiology of Crohn’s Disease (CD), an inflammatory bowel disease (IBD) of humans [3]. MAP and/or CD are also associated with many other autoimmune diseases including multiple sclerosis (MS) [4,5,6,7,8,9,10,11,12,13], type I diabetes mellitus (T1DM) [14,15,16,17,18,19], rheumatoid arthritis [20,21,22], Sjogren’s syndrome [23], asymmetric lateral sclerosis [24], celiac disease [25], depression [26], thyroiditis [27], and the neurodegenerative diseases including Alzheimer’s disease [28] and Parkinson’s disease [29]. A diarrheal/wasting illness associated with infection with MAP has also been reported in non-human primates [30].

Viable MAP is present in our food, potable water and can be isolated from commercially available pasteurized milk [31,32] with MAP being identified in 2.7% of retail pasteurized milk samples purchased in Wisconsin, Minnesota and California, USA [31]. It is highly likely therefore that humans are regularly exposed to this animal pathogen. Screening of human sera with a MAP specific antibody assay has found evidence of MAP specific immune recognition in subjects with a range of underlying diseases including patients with CD and asymptomatic controls [33]. Previous meta-analyses looking predominantly at molecular detection methods also indicated a trend for studies showing a significantly higher percentage of MAP detection in samples from patients with CD, compared to non IBD controls [34,35]. Whilst MAP is undeniably a pathogen in most animal species, efforts to establish a causal relationship between presence of viable MAP infection and similar diseases in humans have been hampered by methods unable to culture MAP strains from human samples in a timely and reproducible manner. Individual smaller studies have cultured MAP from human blood including significantly greater success in CD patients than from controls [36,37]. The method used (MGIT ParaTB culture tubes) was not optimal however, requiring three to six months of incubation and being of relatively low reproducibility. In accordance with consensus recommendations from the MAP conference held at Temple University in March 2017, this study was undertaken to examine alternative approaches able to detect, isolate and/or culture MAP from human blood samples [38].

The primary goals of the study were to test new rapid culture methods for MAP, to assess the degree of viable culturable MAP bacteremia in a test human population, and to determine the relative predominance of MAP in CD patients compared to controls, which included non-CD subjects with various autoimmune diseases. A secondary goal included parallel evaluation of two culture-based approaches developed in different laboratories (TiKa culture and Pozzato culture), a rapid phage amplification detection method, the existing MGIT culture method, and a series of MAP specific antibody tests developed at several other laboratories. Lastly to establish a degree of reliability in the results, selected subjects showing initial positive MAP detection were also followed up with a repeat test one year later.

## 2. Materials and Methods

### 2.1. Study Design and Participants

The study protocol was reviewed on 20 October 2017 by the Temple University IRB (IRB protocol # 24790). This case-control study included 201 subjects—61 patients with CD and 140 non-CD controls. The non-CD control group included 16 patients with UC and various other autoimmune diseases. Within the non-CD control group, there were 24 subjects with arthritis, 17 with hypothyroidism, 9 with UC only, 5 with a combination of UC and other autoimmune diseases, 7 healthy subjects without known disease, 6 with psoriasis, 3 with T1DM, 3 with irritable bowel syndrome (IBS), 2 with rosacea, 1 with asthma, 1 with MS, and 10 with various combinations of the foregoing non-CD, non-UC diseases. The first 159 subjects in the study were recruited from the practice of a gastroenterologist in Winter Park, Florida, Dr. Ira Shafran. In addition, 42 of the subjects were recruited from the Human Paratuberculosis Foundation website (www.humanpara.org) and the phlebotomy of the second group was performed at a site in New York City. A single blood collection was performed on 192 subjects during May through August 2018. For 9 of the subjects a second blood sample was obtained in August 2019 to determine if they had a transient or persistent MAP infection. Selection of these subjects was based on an initial positive phage assay (4/9) or MAP culture (5/9) and for their availability to donate a second sample one year later in Philadelphia. One of these subjects had chronic thyroiditis and IBS, three had chronic thyroiditis, four subjects were asymptomatic and healthy, and one had IBS.

### 2.2. Diagnosis and Diagnostic Categorization

The subjects completed a consent form and a questionnaire about their medical history and BCG status. The patients with CD or UC also completed an additional questionnaire during enrollment to assess the modified Harvey Bradshaw index (HBI). All subject information was kept confidential in accordance with standard medical practice.

### 2.3. Procedures

Blood samples were collected from all 201 participants at enrolment in EDTA blood collection tubes. Peripheral blood leukocytes (PBLs)/buffy coat specimens were prepared from the whole blood samples and then shipped by courier to each of the laboratories performing the cultures (Bull and Grant laboratories in London and Belfast, UK, respectively, and Naser laboratory in Florida, USA). The courier delivery time to the Bull and Grant laboratories was subject to inconsistently applied international customs regulations and thus ranged from 3 to 7 days while the courier delivery time to the Naser laboratory was one day. The plasma from the blood samples was collected and frozen at −80 °C until serology testing was performed later. All samples were identified by a study number and the laboratory scientists were blinded to all clinical information, diagnoses, and personal identifiers.

#### 2.3.1. MGIT Culture

PBL samples from the EDTA buffy coats were inoculated into BACTEC MGIT ParaTB medium with supplements (OADC and mycobactin J as detailed above) and incubated for 6 months at 37 °C. After incubation, the MGIT culture was centrifuged, DNA extracted from the pellet, and nested IS900 PCR was performed as described [36]. Subcultures were made on all PCR-positive MGIT cultures to attempt recovery of MAP in pure culture.

#### 2.3.2. TiKa Culture

Buffy coats transported in Middlebrook 7H9 transport media were centrifuged at 800× *g* for 10 min at RT and the pellet re-suspended in 10 mL freshly prepared TiKa-KiC (TiKa Diagnostics Ltd., London, UK) [39] decontamination cocktail then incubated at 37 °C for 20–24 h with shaking (150 rpm). Samples were centrifuged at 2500× *g* for 15 min at RT, pellets re-suspended in 1 mL recovery medium consisting of Middlebrook 7H9 (Difco, Detroit, MI, USA), plus supplemented casitone (0.1% *w*/*v*), glycerol (0.25% *v*/*v*; Sigma, Merck, Darmstad, Germany), OADC (10% *v*/*v*; Sigma), Mycobactin J (2 μg/mL; ID-Vet, Grabels, France), TiKa14D D-enantiomer peptide WKIVFWWRR (1 μg/mL; TiKa Diagnostics Ltd.), 5-amino salicylic acid (5 μg/mL; Sigma), phylloquinone (18 μg/mL; Sigma) and menaquinone (1.7 µg/mL; Sigma, Merck, Darmstad, Germany) then incubated 37 °C for 2 days. Samples were then added to MGIT culture tubes supplemented with PANTA plus Mycobactin J (2 µg/mL) and TiKa14D (1 µg/mL) and incubated at 37 °C for up to 4 months. Cultures showing visible growth were centrifuged at 2500× *g* for 10 min at RT and pellets subcultured onto solid Pozzato culture (1.5% agar) in 24 well plates then overlaid with semi-solid Pozzato culture 0.75% agar containing TiKa14D (1 μg/mL), sealed with gas permeable membrane and incubated at 37 °C in 5% CO_2_. MAP DNA was extracted from colonies as previously described [40]. Samples were deemed MAP positive if both IS900 and F57 PCR positive.

#### 2.3.3. Pozzato Culture and Phage Amplification Assay

PBL samples were centrifuged (2500× *g* for 15 min) and resuspended in 1 mL Middlebrook 7H9 broth supplemented with 10% OADC (both Difco) and 2 mM CaCl_2_ (Sigma, Merck, Darmstad, Germany). The phage amplification assay and Pozzato culture were performed as described previously [41,42,43,44]. Briefly, following a 15 min incubation at RT and a thorough vortex, 500 µL of each PBMC sample was inoculated into a screw cap glass culture tube containing 4 mL modified 7H9 medium, PANTA antibiotic supplement, and mycobactin J, as described by Pozzato et al. [42] but with no egg yolk added (referred to as” Pozzato culture”) [43]. The second 500 µL of each PBL sample was subjected to the optimized phage amplification assay [44], which proceeded as follows: 10^8^ D29 mycobacteriophages were added to each 1 mL test sample to infect any MAP cells present, and samples were incubated at 37 °C. After 2 h, the extraneous seed phages were inactivated by treatment with virucide (final concentration 10 mM ferrous ammonium sulphate) for 10 min, and then the sample was diluted with 5 mL 7H9/OADC/CaCl_2_ (2 mM) broth. Incubation of samples proceeded until a total of 3.5 h had elapsed since addition of phages, at which point the entire sample was plated with *Mycobacterium smegmatis* mc^2^ 155 sensor cells and 5 mL molten Middlebrook 7H9 agar in Petri dishes. Once solidified, agar plates were incubated overnight at 37 °C and examined next day for evidence of zones of clearing (“plaques”), the presence of which would be indicative of the viable mycobacteria in the sample. In order to confirm the presence of MAP DNA within plaques, a random selection of 10 plaques were excised, DNA extracted and screened for IS900 by PCR as previously described [44]. A PBL sample was deemed MAP culture positive if IS900 and F57 PCR positive biomass (broth pellet or colony on Herrold’s egg yolk-mycobactin J (HEYM) was obtained.

#### 2.3.4. MAP Antibody Assay

Plasma samples for each patient were assayed using the IDEXX *Mycobacterium paratuberculosis* antibody test kit for detection of antibody to MAP in bovine serum, plasma, and milk. This kit was adapted for human use as described previously [45]. Human plasma controls optical density (OD) values were used to calculate sample/positive (S/P) ratios and interpret the assay.

#### 2.3.5. PtpA and PknG ELISA Test

The antibodies against virulence factors secreted by MAP during infections were measured in plasma specimens as described previously [46,47]. These antigens included the protein tyrosine phosphatase PtpA and the protein kinase G PknG [47,48]. Recombinant PtpA and PknG were produced in *M. smegmatis* and according to published procedures [47,48].

#### 2.3.6. Hsp65 Antibody Assay

Hsp65 antibody from the blood was measured by direct ELISA assays described previously [23]. Recombinant Hsp65 from *Mycobacterium avium* subspecies *hominissuis* (MAH) was produced at GenScript Corp (https://www.genscript.com/) through contract work and used to coat the 96-well plate.

### 2.4. Data Analyses and Statistical Methods

Data were expressed as frequencies and percentages for categorical variables and mean ± standard deviation (SD) and/or median (range or quartile range) for continuous variables. Associations between potential risk factors or assay methods of interest and select disease status (i.e., CD or CD + UC) were evaluated using the Fisher’s exact test for two groups and Wald test for a continuous variable. Multivariable logistic regression analyses were performed on the select diseases to explore their association with or predictability of the disease using different assay methods (culture, antibodies or both) when other potential risk factors or confounding variables such as age and gender are adjusted in the regression model. To define a cutoff value of a continuous covariate or an assay variable for inclusion in a logistic regression model in predicting a patient’s outcome, the cutoff was chosen to achieve an optimal classification criterion based on the Euclidean distance method. Both the continuous and dichotomized versions of continuous variables were included in the logistic regression model as candidate variables in a stepwise variable selection procedure. A stay probability of 0.10 and entry probability of 0.25 were employed to arrive at a final regression model. Therefore, variables with non-significant predictive abilities for a disease state were dropped from the multivariable logistic regression model to keep the model parsimonious. Age and sex were included in all regression models a priori. Both raw and adjusted odds ratios of having CD or CD + UC and their 95% confidence intervals (CIs) were reported whenever appropriate. *p*-values less than 0.05 were considered statistically significant. SAS version 9.4 (SAS Institute Inc., Cary, NC, USA) was used for all the data analyses.

## 3. Results

The subject demographic data by subgroups for the study appears in Table 1. The median patient age and age ranges are presented in addition to the mean age and standard deviation because the data is asymmetric, and this approach is favored for asymmetric distributions such as age data. In the study, the non-CD patients appeared to be slightly older than the CD patients (median: 57.5 vs. 47 years old) and have a similar gender composition (59% vs. 54% females).

The analytical sensitivities (the ability of the method to detect the organism, rather than clinical sensitivity, which is the ability of the test to detect disease) of the culture methods for viable MAP bacteremia in order of the highest to lowest are: (1) Pozzato culture (124/201 or 62% of all subjects, 35/61 or 57% of CD patients), (2) Phage assay (113/201 or 56% of all subjects, 28/61 or 46% of CD patients), (3) TiKa culture (64/201 or 32% of all subjects, 22/61 or 36% of CD patients) and (4) MGIT culture (36/201 or 18% of all subjects, 15/61 or 25% of CD patients). These results are summarized in Table 2.

All positive MAP cultures reported from each of the three culture methods and the interrelationships between results of the three liquid culture methods (MGIT, TiKa and Pozzato) are illustrated in Figure 1. Appendix A show numbers of plaques observed in CD patients and non-CD controls (Appendix A).

Table 3 depicts the association of clinically diagnosed CD patients with various categorical variables of interest. Of note, the younger age group (≤52 years old) seemed to be more likely to have CD when compared to the older age group (>52 years old) (OR (95% CI): 2.66 (1.42, 4.99); *p* = 0.003). Among all assay methods, a negative Hsp65 antibody assay <0.74 appeared to differentiate CD from non-CD cases (*p* = 0.06 and 0.02, see Table 3).

Table 4 contains the percentages of subjects showing evidence of MAP infection by all of the cultural methods used in the study. Viable MAP bacteremia was also detected in patients with autoimmune conditions who did not have either CD or UC: (1) Pozzato culture (33/57 or 58%), (2) Phage assay (36/57 or 63%), (3) TiKa culture (19/56 or 34%) and (4) MGIT culture (9/57 or 16%). Within this group, there were 24 subjects with arthritis, 17 with hypothyroidism, 6 with psoriasis, 3 with T1DM, 3 with IBS, 2 with rosacea, 1 with asthma, and 1 with MS. All 9 subjects who had an initial positive phage (Figure 2) assay (4/9) or MAP culture (5/9) were positive on the second Phage assay from a second blood sample obtained one year later.

To investigate the independent relationships of each of the culture and serologic methods with the presence of MAP and the clinical diagnosis in the subjects, i.e., CD or non-CD, a multivariable logistic regression model adjusted for age and gender was used for association analyses and this data appears in Table 5. The significant findings regarding associations with MAP culture and/or antibody data are provided using the odds ratio (OR) and its 95% confidence interval (CI) for having CD or having either CD or UC and a *p*-value. The OR data for those culture and serologic methods that achieved significance is presented adjusted for age and gender (Table 5), was found to be more robust than the unadjusted/marginal odds ratio data (Table 3).

The associated *p*-value was 0.037 with an OR (95% CI) of 2.36 (1.06, 5.28) for the MGIT culture for MAP positivity corresponding to CD patients and this associated *p*-value supports the predictive power of this culture method. This was also true when UC subjects were included along with the CD subjects in the analysis and the result of this analysis was a *p*-value of 0.006 and adjusted OR (95% CI) of 3.19 (1.40, 7.23) for having CD + UC when comparing MGIT culture positive to negative subjects.

The TiKa culture showed a significant association (data not included) for positive MAP cultures, but the TiKa, Phage assay and Pozzato culture methods did not reach the statistical significance at 0.10. Therefore, the preceding methods did not make the final model predicting the presence of CD in our study. Nonetheless, the Phage assay and Pozzato culture methods detected a higher proportion of MAP infections in the non-CD/UC cohort while TiKa culture had a reverse trend similar to the one exhibited by the MGIT culture method (see Table 2).

Amongst the several serology tests, the highest and only significant correlation to the presence of CD occurred with the Hsp65 antibody. At a cutoff value of 0.74, this method had the best ability to discriminate between CD patients and non-CD subjects (adjusted OR (95% CI) of having CD comparing Hsp65 Ab > 0.74 vs. Hsp65 Ab ≤ 0.74: 2.40 (1.25, 4.61); *p*-value 0.009; Table 5). The PknG antibody also had a significant negative correlation to the presence of CD/UC in our patients as compared to the non-CD/UC subjects (adjusted OR (95% CI) of having CD/UC comparing PknG negatives to positives: 2.18 (1.12, 4.23); *p*-value: 0.022. Spearman correlation showed that the Hsp65 antibody had a weak correlation with the HBI (Spearman coefficient = 0.28, *p* = 0.03) and the Phage Plaque counts showed a very weak correlation with the HBI (Spearman coefficient = 0.12, *p* = 0.37) among 60 CD patients who had such data. It is worth noting that the range of the HBI in the study population was limited and few patients had an HBI greater than 5.

The MGIT culture and the Hsp65 antibody assay were the best discriminators between the CD patients and the non-CD subjects for the study population, both each in its individual own (culture or antibody) category and together independently in the culture and antibody combined set, after having adjusted for age and sex (and each other in the latter case) (see Table 5). Table 6 shows the agreement analysis between the Phage assay and each of the other MAP culture methods. The greatest percentage agreement (58%) occurred between the Phage and the Pozzato methods performed in the same lab on the same PBL sample, whereas the percentage agreement between the Phage and the other two culture methods was similar but diminished (TiKa 47% and MGIT 48% agreement).

## 4. Discussion

This study demonstrates viable MAP organisms to be present in the blood of a significantly high number of humans and that this state of infection can be persistent. Viable MAP bacteremia was found not exclusive to any one of the groups tested including patients with CD, UC, CD with UC, a range of other autoimmune diseases or asymptomatic subjects.

The study has compared three culture methods and a phage based plus culture method, tested blinded and in parallel on aliquoted samples of blood buffy coats processed at differing laboratories with expertise in using each method. All positive MAP cultures were identified and confirmed by validated specific molecular identification using either nested IS900PCR, ([36]) or IS900 plus specific MAP gene F57 PCRs [40,49]. Aliquots of the same samples were studied by each of the methods, but sufficient funding was not available to perform the same methods in all laboratories. The methods selected included a MGIT culture method previously used for MAP in human PBLs advocated by Naser [37] and two new culture methods; TiKa decontamination and culture [39] and culture in 7H9+ broth (herein referred to as Pozzato culture) [42,43]. Culture of MAP from buffy coat white cell fractions using the MGIT method alone showed an increased correlation with CD patients compared to non-CD controls. The other two methods (TiKa and Pozzato culture), whilst also culturing MAP from a significant number of patients did not yield a significant difference between any of the groups. Previous meta-analyses (Feller and Abubakar, 2007) [35] have indicated a higher rate of MAP detection, using molecular detection methodology in samples from CD patients compared to controls.

Our culture results as a whole do not fully support this conclusion; however, these previous analyses were predominantly based on data from randomly targeted gastrointestinal mucosal biopsies, whilst our study tested blood samples, which could account for the discrepancy. In addition, and again in parallel on blinded aliquots of samples at a separate laboratory, we included an optimised phage amplification assay designed to detect viable MAP [44]. This rapid culture-based/phage amplification assay detects only viable mycobacterial cells present in samples that are able to take up and amplify mycobacterial specific phages, which then burst to release progeny phages. When released these can be plaque assayed and the plaques (containing the original lysed mycobacteria) subjected to species specific PCR to detect and quantify MAP presence within 48 h. In this study aliquots of the phage tested samples were in addition incubated in Pozzato medium and re-tested to confirm positivity after a period of growth. This method was able to detect viable mycobacteria in samples within 2 days and importantly was also able to show that of nine patients in our study that were initially MAP positive and could be followed up after one year, all remained positive for MAP in their blood after this time.

The results show that both alternative culture approaches (TiKa and Pozzato cultures) were able to culture viable MAP more effectively than the reference MGIT method, irrespective of subject group. These findings suggest that the composition of MGIT ParaTB medium is not optimal for isolation of MAP from human PBLs. Whilst the MGIT culture approach may have yielded a greater number of MAP positive cultures from PBLs of CD patients compared to non-CD controls (the expected outcome in light of previous human PBL testing of CD patients [36,37]), a similar or higher number of non-CD controls tested positive for viable MAP than CD patients by the TiKa and Pozzato culture methods. One explanation for this observation is that MAP phenotypes, persistent in blood are not in a fully growth competent physiological state or as occurs in other mycobacterial species have entered dormancy phases. This is possibly more particularly so in those patients receiving medication/chemotherapy which could have been involved in inducing these phenotypes. Hence some form of resuscitation of MAP may be necessary before growth can occur. The differing composition of TiKa and Pozzato broths could offer a reason for this discrepancy in MAP resuscitation capability. Both are based on Middlebrook 7H9 but the TiKa system includes TiKa supplemented Pozzato culture medium followed by long term culture in TiKa supplemented MGIT culture whilst the Pozzato medium used in our study excluded the addition of egg yolk as previously described to isolate MAP from cattle faeces [42] to allow optical density of cultures to be monitored during incubation. Further work is necessary to determine the essential ingredient in this case.

A further difference between sample testing in each of the laboratories was the age of the PBLs at the time of culture. Sample transport times differed which could have affected the conditions to which they were exposed. It is plausible the viability of MAP cells in the PBL samples may have reduced the longer it took for transit. This is evidenced by significant differences in plaque numbers obtained with the phage assay for PBL samples that were older at the time of testing (Appendix A). Perhaps it is beneficial to delay PBL testing for a few days to allow either resuscitation of MAP cells in the transport medium (which was MGIT medium with OADC added) or lysis of PBL cells making MAP cells more available for culture.

Our results indicate a significant percentage of subjects had MAP bacteremia, irrespective of the underlying disease and positive in more than one of the culture methods. There are three possible explanations for this finding: (1) viable MAP is passively acquired from consumed food and not immediately cleared by the host following food consumption. This explanation is unlikely because other organisms that are consumed in the diet, survive the digestive process, and cross the “leaky bowel” in IBD into the blood do not cause persistent viable bacteremia in these hosts; (2) MAP is present, persistent and viable but does not cause disease in any human host. This explanation is also unlikely since there are no previously known examples of persistent bacteremia by a known pathogen without conferring some effect (3) MAP persistently infects some hosts and as a pathogen is able to influence disease processes in some susceptible human hosts. We believe that the third explanation is most likely, and that MAP represents a zoonotic agent. In this case human infection resembles the known pathogenic state in cattle in which only a minority of animals (10%) develop advanced clinical JD with the majority exhibiting sub-clinical persistent infection [50].

The following caveats apply to this study. These statistics were based on a sample population that was not a random sample nor guaranteed to be representative for age, gender, disease severity or health. In addition, neither the Hsp65 antibody nor the MAP IDEXX antibody assay are specific for MAP and may cross react with other mycobacterial species. This study points to the urgent need for further in-depth MAP studies to fully explore the role of this organism in humans. An obvious area of further research is the discovery of better therapies applied in controlled clinical trials that target and attempt to eliminate MAP [51]. MAP culture and phage assay studies should be conducted in other diseases of unknown etiology by experts of these methods. Finally, as a minimum measure of best practice, the possibility that MAP is a zoonotic pathogen should prompt public health measures to better control JD and MAP spread into food and the environment by governments worldwide.

## Figures and Tables

**Figure 1 microorganisms-08-02054-f001:**
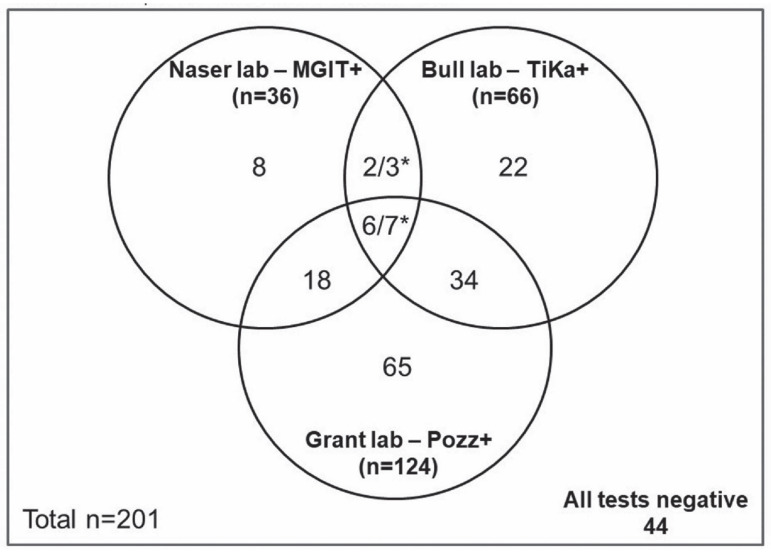
All positive MAP cultures from each of the culture methods and the interrelationships between the three methods. * Two samples from Bull’s lab were discarded from analysis due to non-mycobacterial contamination of media during sub-culture.

**Figure 2 microorganisms-08-02054-f002:**
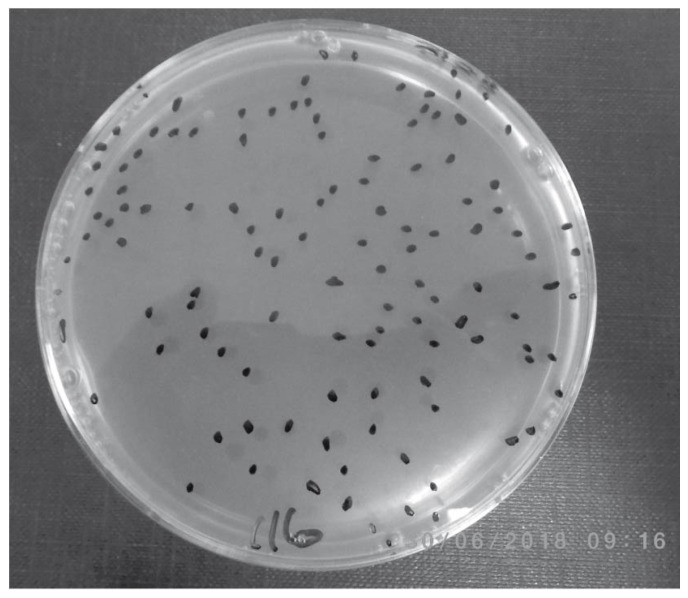
Representative Phage assay result for a MAP positive PBL sample; in this case showing 116 plaques.

**Table 1 microorganisms-08-02054-t001:** Demographics and Clinical Characteristics by Subgroups (*n* = 201).

Variable	CD Patients (2 also Had UC) (*n* = 61)	UC Patients only (*n* = 14)	Non-CD Patients (Including UC only Subjects) (*n* = 140)
Age, N	61	13	130
Mean (SD)	46.2 (15.5)	44.1 (17.3)	52.6 (17.4)
Median (Range)	47.0 (21.0–72.0)	42.0 (24.0–80.0)	57.5 (17.0–85.0)
Gender, N (%)			
Female (%)	33 (54.1%)	7 (50.0%)	82 (58.6%)
Male (%)	28 (45.9%)	7 (50.0%)	58 (41.4%)
HB Index, N	60	13	NA
Mean (SD)	2.4 (2.0)	1.5 (1.8)	NA
Median (Range)	2.0 (0.0–7.0)	1.0 (0.0–5.0)	NA

**Table 2 microorganisms-08-02054-t002:** Analytical sensitivity of MAP detection by phage assay and MGIT, TiKa and Pozzato culture methods among 201 study subjects.

Subject Category	Detection Method			
	Phage Assay	Pozzato Culture	TiKa Culture ^†^	MGIT Culture
CD *(*n* = 61)	28 (46%)	35 (57%)	22 (36%)	15 (25%)
UC only (*n* = 14)	6 (43%)	10 (71%)	1 (7%)	4 (29%)
Non-CD ** (*n* = 140)	85 (61%)	89 (64%)	42 (30%)	21 (15%)
All subjects (*n* = 201)	113 (56%)	124 (62%)	64 (32%)	36 (18%)

Results are reported as number (%) of subjects in each category, with subject numbers in parentheses for CD and UC only patients. * Two CD patients also had a diagnosis of UC. ** Non-CD category includes the 14 UC only subjects. ^†^ Two subjects had missing TiKa culture results.

**Table 3 microorganisms-08-02054-t003:** Association analyses of clinically diagnosed CD patients with covariates of interest.

Variable	N	CD Patients (*n* = 61)	Non-CD Patients (*n* = 140)	Odds Ratio	Fisher’s Exact *p*-Value
Age, overall	191	61 (31.9%)	130 (68.1%)		0.003
≤52	91	39 (42.9%)	52 (57.1%)	2.66 (1.42, 4.99)	
>52	100	22 (22.0%)	78 (78.0%)	Reference	
Gender, overall	201	61 (30.3%)	140 (69.7%)		0.64
Male	86	28 (32.6%)	58 (67.4%)	1.20 (0.65, 2.20)	
Female	115	33 (28.7%)	82 (71.3%)	Reference	
MGIT Culture, overall	201	61 (30.3%)	140 (69.7%)		0.11
Positive	36	15 (41.7%)	21 (58.3%)	1.85 (0.88, 3.89)	
Negative	165	46 (27.9%)	119 (72.1%)	Reference	
TiKa Culture, overall	199	60 (30.2%)	139 (69.8%)		0.41
Positive	64	22 (34.4%)	42 (65.6%)	1.34 (0.71, 2.53)	
Negative	135	38 (28.1%)	97 (71.9%)	Reference	
Pozzato Culture, overall	201	61 (30.3%)	140 (69.7%)		0.43
Negative	77	26 (33.8%)	51 (66.2%)	1.30 (0.70, 2.39)	
Positive	124	35 (28.2%)	89 (71.8%)	Reference	
Phage Assay, overall	201	61 (30.3%)	140 (69.7%)		0.064
Negative	88	33 (37.5%)	55 (62.5%)	1.82 (0.99, 3.34)	
Positive	113	28 (24.8%)	85 (75.2%)	Reference	
Hsp65 Antibody, overall	201	61 (30.3%)	140 (69.7%)		0.020
>0.74	89	35 (39.3%)	54 (60.7%)	2.14 (1.16, 3.95)	
≤0.74	112	26 (23.2%)	86 (76.8%)	Reference	
PknG Antibody, overall	200	61 (30.5%)	139 (69.5%)		0.35
Negative	120	40 (33.3%)	80 (66.7%)	1.40 (0.75, 2.63)	
Positive	80	21 (26.3%)	59 (73.8%)	Reference	
PtpA Antibody, overall	200	61 (30.5%)	139 (69.5%)		0.73
Negative	146	46 (31.5%)	100 (68.5%)	1.20 (0.60, 2.39)	
Positive	54	15 (27.8%)	39 (72.2%)	Reference	
MAP IDEXX Ab, overall	201	61 (30.3%)	140 (69.7%)		0.62
Positive	135	43 (31.9%)	92 (68.1%)	1.25 (0.65, 2.39)	
Negative	66	18 (27.3%)	48 (72.7%)	Reference	

**Table 4 microorganisms-08-02054-t004:** Comparison of the number and percentages of subjects showing evidence of MAP detected by all cultural and serological methods employed; separately for CD patients, UC only patients and Non-CD controls (including 14 UC only patients).

	Lab/Detection Method							
Subject Category	Naser/ MGIT Culture	Bull/ TiKa Culture	Grant/ Pozzato Culture	Grant/ Phage Assay	Potula/ IDEXX ELISA	Bach/ PtpA	Bach/ PknG	Zhang/ Hsp65
CD patients (*n* = 61)	15 (24.6%)	22 (36.1%)	35 (57.4%)	28 (45.9%)	43 (70.5%)	14 (23.0%)	21 (34.4%)	39 (63.6)
UC patients (*n* = 14)	4 (28.6%)	1 (7.1%)	10 (71.4%)	6 (42.8%)	8 (57.1%)	4 (28.6%)	3 (21.4%)	6 (42.8%)
Non-CD or UC with autoimmune condition (*n* = 58)	9 (15.5%)	19 (32.7%)	34 (58.6%)	36 (62.0%)	40 (68.9%)	12 (20.7%)	30 (51.7%)	23 (39.6%)
Non-CD, non-UC and non-autoimmune condition (*n* = 68)	8 (11.8%)	22 (32.3%)	44 (64.7%)	43 (63.2%)	44 (64.7%)	23 (11.8%)	26 (38.2%)	26 (64.7%)

**Table 5 microorganisms-08-02054-t005:** Association analyses of MAP culture and serology with occurrence of select disease (CD or CD + UC) ^†^.

Assay Method	CD vs. Non-CD			CD + UC vs. Non-CD + UC		
	*n*	Adjusted Odds Ratio (95% CI)	*p*-Value	*n*	Adjusted Odds Ratio (95% CI)	*p*-Value
Using Culture only	191			191		
MGIT Culture (+ vs. −)		2.36 (1.06, 5.28)	0.037		3.19 (1.40, 7.23)	0.006
Using Antibody only	191			190		
Hsp65 Antibody (>0.74 vs. ≤0.74)		2.40 (1.25, 4.61)	0.009		1.32 (1.05, 1.67) ^‡^	0.016
PknG Antibody (− vs. +)			NS		2.18 (1.12, 4.23)	0.022
Using both Culture and Antibody	191			190		
MGIT Culture (+ vs. −)		2.54 (1.11, 5.81)	0.027		3.51 (1.51, 8.16)	0.004
Hsp65 Antibody (>0.74 vs. ≤0.74)		2.51 (1.29, 4.88)	0.007		2.30 (1.19, 4.45)	0.013
PknG Antibody (− vs. +)			NS		2.13 (1.08, 4.20)	0.030

^†^ Results reported from six (6) logistic regression models (3 assay method variables inclusion settings × 2 select diseases (CD or CD + UC), all adjusted for age (≤52 vs. >52) and gender; NS: not significant. ^‡^ OR and 95% CI for a 0.2-unit increment of Hsp65 antibody.

**Table 6 microorganisms-08-02054-t006:** Agreement analysis between Phage assay and culture results (*n* = 201).

Phage Assay Result	Comparator Test Results		Total	Agreement between Tests
			N (%)	%
	Pozzato +	Pozzato −		
Phage assay +	76 (38%)	37 (18%)	113 (56%)	58%
Phage assay −	48 (24%)	40 (20%)	88 (44%)	
Total N (%)	124 (62%)	77 (38%)	201	
	TiKa +	TiKa −		
Phage assay +	35 (18%)	76 (38%)	111 (56%)	47%
Phage assay −	29 (14%)	59 (30%)	88 (44%)	
Total N (%)	64 (32%)	135 (68%)	199 *	
	MGIT +	MGIT −		
Phage assay +	22 (11%)	91 (45%)	113 (56%)	48%
Phage assay −	14 (7%)	74 (37%)	88 (44%)	
Total N (%)	36 (18%)	165 (82%)	201	
	Any culture +	All culture −		
Phage assay +	91 (45%)	22 (11%)	113 (56%)	56%
Phage assay −	66 (33%)	22 (11%)	88 (44%)	
Total N (%)	157 (78%)	44 (22%)	201	

***** Two samples from Bull’s lab were discarded from analysis due to non-mycobacterial contamination of media during sub-culture.

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
