# Peer review of "Presence of Infection by Mycobacterium avium subsp. paratuberculosis in the Blood of Patients with Crohn’s Disease and Control Subjects Shown by Multiple Laboratory Culture and Antibody Methods"

_microorganisms, 2020, doi:10.3390/microorganisms8122054_

Round 1
Reviewer 1 Report
Thank you for the clarification!
Author Response
Thank you!
Reviewer 2 Report
This is the revised version of a manuscript which I reviewed in the past. The Authors satisfactorily addressed my concerns.
Author Response
Thank you
Reviewer 3 Report
This is a much improved version of the paper. There are a few more issues with regards to how the study was done and data analysis that need to be clarified.
1. Intro, 2nd paragraph - I am not sure I understand how the lack of a reliable culturing method has hampered any mechanistic studies. If that were the case, this study also does not address it adequately due to a flaw in the design and how the samples were handled.
2. Table 1, Were these variables (age, gender, etc.) adjusted for in the analysis by culture method? What type of analysis other than descriptive parameters were performed?
3. Table 2 - while analytical sensitivity is a good parameter to have, in a study that aims at comparing multiple methods a "Kappa" value or "percent agreement" between them would be needed. This assumes that bacteremia is present in all CD patients. This is in and of itself a major concern. The denominators are likely incorrect. This needs to be clarified and explained or taken out from the paper.
4. Figure 1 - Suggests low or no agreement between methods. This is why multiple method evaluation should include all tests by all labs. This way, an inter lab variability or inter-method reliability can be established. As done, it is impossible to tell if it is the method or the lab doing the test is the reason for low agreement.
5. Discussion, 2nd paragraph,
Phages are not MAP-specific and need to be followed up with a PCR for confirmation. While this is stated in discussion no data is provided for the phage assay performed across all three methods. If this was done, it should be clarified.
Author Response
RESPONSE TO REVIEWERS
Reviewer 3:
Comment 1. Intro, 2nd paragraph - I am not sure I understand how the lack of a reliable culturing method has hampered any mechanistic studies. If that were the case, this study also does not address it adequately due to a flaw in the design and how the samples were handled.
Response. This section has been revised to appropriately convey the author’s intent.
Comment 2. Table 1, Were these variables (age, gender, etc.) adjusted for in the analysis by culture method? What type of analysis other than descriptive parameters were performed?
Response.The variables of age, gender, etc. were not adjusted for in the analysis by culture method.
Comment 3. Table 2 - while analytical sensitivity is a good parameter to have, in a study that aims at comparing multiple methods a "Kappa" value or "percent agreement" between them would be needed. This assumes that bacteremia is present in all CD patients. This is in and of itself a major concern. The denominators are likely incorrect. This needs to be clarified and explained or taken out from the paper.
Response.We appreciate reviewer’s comment. The section describing "Kappa" value in the results section had been revised. Table 6 is also revised for clarity.
Comment 4. Figure 1 - Suggests low or no agreement between methods. This is why multiple method evaluation should include all tests by all labs. This way, an inter lab variability or inter-method reliability can be established. As done, it is impossible to tell if it is the method or the lab doing the test is the reason for low agreement.
Response.This point was addressed in the revised manuscript in the discussion section, 2ndparagraph. “Aliquots of the same samples were studied by each of the methods, but sufficient funding was not available to perform the same methods in all laboratories.
Comment 5. Discussion, 2nd paragraph. Phages are not MAP-specific and need to be followed up with a PCR for confirmation. While this is stated in discussion no data is provided for the phage assay performed across all three methods. If this was done, it should be clarified.
Response. As suggest this section has been further clarified in the current version of the revised manuscript.
Round 2
Reviewer 3 Report
Authors have addressed all comments.
This manuscript is a resubmission of an earlier submission. The following is a list of the peer review reports and author responses from that submission.
Round 1
Reviewer 1 Report
This study performed 3 distinct bacterial cultures and a phage lysis assay in 3 different labs on blood (PBMCs) of patients with CD, UC, or CD+UC and controls. While the study generated data, the design itself is insufficient to make the types of comparisons that the authors present. This is evident in the data as well.
It is unclear why a genome sequence was thrown into the manuscript (not in the initial aims of the study as presented) diluting the impacts of an association study. It is unclear as to what to make of the genome sequence data presented - so it is about 40+ SNPs separated from commonly isolated cow strains, so what is the significance and how does it correlate with the culturing methods used? Or is that not the point? If not, how please explain. Was this isolated from all 3 culturing methods? Did each method have the same genome sequence? Too many side questions arise, I suggest removing the genome sequence data and presenting that for the (six or seven) isolates that came out from all 3 methods. If the authors want to keep the genome sequence data, the must deposit these in the public domain to enable comparative analyses. I strongly suggest taking it out of this paper and designing a better comparative genomic analysis for another study. Sequencing more genomes are not necessarily better but with appropriate designs and study questions, it can be quite impactful. it is not, in this study.
Other major points -
Abstract: Should be rewritten to help the reader clearly understand the study and the study design.
Line 5 - Incorrect use of "infection" It may be better to state "... the higher rate of MAP presence or bacteremia..."?
Line 9 - n = 201 or 217? I counted 217 (CD+UC, CD alone, UC alone' and controls) - 2+61+14+140 = 217
Unclear here - it is better worded in the Methods section. Also, among the controls how many were devoid of any co-morbidities or other autoimmune conditions? Need this in methods as well.
Need the OR data here as well, despite the fact that it is not significant by lab/method.
Line 17 - calls out 9 phage assay positives (or 1 culture-positive sample as being positive a year later. Was that 1 culture a subset of the 9? Were these 9 detected as culture-positive by any method? Unclear as presented.
Introduction -
Second paragraph - It is actually not the "association" that is controversial. It is rather the mechanism that explains the disease process that is lacking - needs an unequivocal demonstration of causation. So, I request that this sentence be reworded.
Last paragraph, last sentence - how were these "selected" for retesting a year later?
Methods -
1. Were any health controls (with no other autoimmune condition) included?
2. Were PCR products (IS900 or the F57) confirmed by sequencing?
3. Section 2.3.3 starts with PBMCs but a previous section states PBL. Please harmonize the entire document to use one
4. Hsp65 ELISA - How much (concentration per well) protein (hsp65) was used to coat plates? What kind of plates were used? What were the serum dilutions used in testing and where was the secondary antibody sourced? Also was this a colorimetric or chemi or fluor detection method? Details needed.
5. Why were PtpA and PknG ELISAs performed? Again, need details on the quantity of the protein used, detection systems, etc.
6. WGS - I suggest eliminating this section and all allusion to that work. Seems very disjointed and doesn't fit the theme of this work. Also, this section is confusing! Was this from TiKa or Pozzato culture? They start with tiKa positive and say 3 colonies fro Pozz culture were used....what was it?
7. TiKa Culture section -
a. Why is the decontamination step needed for a normally sterile body fluid (like blood)? Will this compromise the detection of MAP?
b. What are these supplements (A, M1, M2, etc.)? It would be near impossible to reproduce or demonstrate the consistency of these findings if the readers don't know the ingredients to use. As such, this method is defunct given that other labs can't reproduce.
Do authors mean colony counts when they refer to "biomass"?
8. MGIT Section - Were PCR products confirmed by Sanger sequencing? This is more important than showing one WGS data.
Table 1. - What proportion of non-CD patients have no co-morbidities (i.e., apparently healthy)? Also, ages of the patients suggest that these subjects had the disease for a duration. Can you be a bit more specific about "time since diagnosis"? Also, if retail milk positivity data is accurate, these subjects have been likely exposed for a fairly long duration in their lifetimes. Is there an additive effect with time? Is it possible that CD/UC/IBS patients may have compromised intestinal barriers leading to higher rates of MAP escape into the bloodstream and a positive culture being of no consequence? What use is the detection of viable bacteremia if that cannot be equivocally established as causation?
Figure 1. - Each lab performed a different assay -this Venn provides little value except that identifies a flaw in the design. If TiKa or Pozz cultures were done by all three labs on the sample sets, then looking at the interlaboratory agreement would be useful - this study design does not allow for this type of comparison as something different was done by each lab. However, it is clear that each lab and method were detecting different samples as positive! This data is quite disturbing as only 6 or 7 were detected by all three labs as positive. This data further emphasizes that this study was not reproducible by the 3 labs/methods to a large extent. This is also clear in the OR data presented in the next table - none of the individual methods was significant. It is not correct to show different methods performed by different labs as a combined OR.
If the interest is to compare culturing methods, all 3 labs should have performed all 3 methods on the same set of blonded samples. So this study design is not amenable to compare culturing methods (data also suggests the same - detection is a function of method/lab). The same goes for serology.
Table 3 - What does 0.74 signify for hsp65 ELISA? Where (whose lab) was this done?
Table 4 - I'd be interested to see hsp65 ELISA versus each Lab culture data.
The paragraph below table 4 - Each culturing method was done by a different lab so cannot rule out laboratory effect. Was this in the logistic regression model as a parameter?
Discussion - first paragraph - Is confusing because each method showed the presence of bacteremia in a separate set of isolates except for 7 that were commonly positive across labs. So making such a strong statement as "Definitively demonstrates" is incorrect! Seven of 201 is not definitive. The method effects or lab effects are not included in the analysis.
Reviewer 2 Report
In this article, Kuenstner and colleagues compare various culture methods, phage assays and immunological tests for the detection of Mycobacterium avium subsp. Paratuberculosis infection in the blood of Crohn’s disease (CD) patients as well as of control subjects. The most sensitive assay seems to be the Pozzato culture, which was able to detect the microorganisms with high frequency in the sample set. However, an immunological assay based on Hsp65 showed the best predictive value in differentiating between CD and non-CD cases.
Experiments and statistical analyses seem to have been conducted properly. However, I have a few concerns that should be addressed:
- I don’t understand where Figure 2 is. I can read the legend to Figure 2 on page 7 but there’s no figure there.
- On page 9 I read that “WGS will be performed on 40 additional MAP culture isolates from this study and reported as an addendum”. This sentence sounds quite unorthodox to me. Data should be provided and analysed in full when a manuscript is submitted to a journal.
- As a general comment, data should be better presented and more elaboration is needed. For instance, Figure 1 should be more extensively described - there is only one line in the text as a comment to what is reported there. On the other hand, if Figure 1 is deemed unnecessary, then it should be removed.
- The legend to Table 2 seems to be part of the main text - it should be placed just below Table 2.
- What's the relevance of the dendrogram in Figure 5? If this Figure adds important information to the message of the manuscript, then it should be presented clearly and discussed, otherwise it should be removed. The same comment applies to Figure 4.
- The Discussion section is too long for what is really here. It should be shortened and, most importantly, it should address the major issue that the Authors report in the Introduction, i.e. “problems and controversy in proving the association of MAP with any of these conditions has been hampered by…reproducible methods”. Also, the Discussion about MAP infection as a zoonosis should be shortened or removed since this point is not the subject of the manuscript.
Reviewer 3 Report
Figure 2 looks more like a table. Figure 1 - please explain what is meant by "contaminated" result. A nested PCR was used for IS900 detection. Given no evidence that measures were taken to inactivate amplicons that might have been introduced into reactions, what is the chance that some results were falsely positive? This may be a fourth possible explanation for the results observed (page 11, 2nd paragraph).